# OpenReview forum: "MetaCog-Bench: A Process-Based Benchmark for Evaluating Metacognitive Monitoring and Control in Large Language Models"
_TMLR — Under review for TMLR_

### Review · Reviewer_fY8v · 2026-05-26

**Summary Of Contributions:**

The manuscript describes a benchmark for evaluating metacognitive abilities of LLMs. Metacognition refers to self-awareness of one's own abilities and acting in ways that take that self-knowledge into account. Existing benchmarks for LLM meta-cognition often use LLM-as-judge, which is scalable but biased. When evaluating a model using a model from the same family as judge, metacognitive gaps will be shared, making the judge unable to detect mistakes. The benchmark presented here instead uses deterministic evaluation based on text matching, e.g. a task is marked as passed if the model's response contains specific words. They test 7 different models. Overall, they find that Grok-3-mini and Deepseek-V3 show the best metacognitive abilities but different models have divergent metacognitive profiles, excelling in different subtasks.

## Strengths:
- Design of benchmark tasks and metrics are theoretically grounded and informed by the cognitive science literature about metacognition
- The analysis of the different metacognitive profiles is interesting
- It is also interesting that asking for the response in JSON affected the scores so much! This might be a generic issue that could affect lots of evaluations that ask for responses in JSON format.

## Weaknesses:
- Without a ground truth, there is no basis on which to compare the deterministic approach used here to LLM based evaluation approaches. As far as a reader can tell, the deterministic approach could be underestimating as much as the LLM-as-judge approach is overestimating.
- The results of the systematic audit are buried in the limitations section.
- No comparison to non-self LLM-as-judge. Is the self-bias resolved by using a judge from a different model family?

**Audience:**

Yes

**Audience Explanation:**

There are many individual results in the paper that will be of interest to a wide audience interested in differences in abilities between models and how to approach LLM evaluation generally, even readers not seeking a new benchmark for metacognition.

**Broader Impact Concerns:**

The broader impact statement is adequate.

**Claims And Evidence:**

No

**Claims Explanation:**

Some claims about the superiority of the proposed deterministic approach are not well supported. For example, the paper reports that the LLM-as-judge approach "inflates scores dramatically" but without a ground truth, it is not clear how inflated the LLM-as-judge response is or whether the deterministic approach is more sensitive.

**Requested Changes:**

## Critical
- You describe a systematic audit of random responses but the details are spread over three sections. Results are found in the abstract ("100 random samples", ">96% accuracy") and in the limitations section ("the false positive rate on the audited sample was <4% and the false negative rate was <8%"). This analysis is super important to interpret your results and should be reported fully in the results section. Please include the false positive and false negative rate before and after the changes you made to the keyword matching criteria and of the LLM-as-judge approach on these examples.

## Non-critical:
- Right now the results are interesting, but the paper does not convince a reader to prefer this benchmark over other approaches.
	- The paper would be much improved if you had a ground truth for subsamples of the other tasks as well, not just Tier 2, and evaluated the LLM-as-judge approach on them.
	- The paper would be much improved if you included a comparison to a non-self LLM-as-judge approach (using a judge from a different model family).

## Minor:
- Plotting the diamonds for confidence (your independent variable) on Figure 3 is confusing. The dashed line along the diagonal is sufficient.
- The individual scores on Figure 5 are illegible
- On page 9 you describe a post-decision bias. Such biases also occur in humans and it would be interesting to mention this in the article. e.g. https://www.sciencedirect.com/science/article/pii/S2352154616300973 or https://www.annualreviews.org/content/journals/10.1146/annurev-psych-022423-032425
- Some of what you test is also a design choice for AI developers. Do they want to release and AI that answers the question as asked, or infers the question that "should" have been asked? This might be less about metacognitive ability and more about what companies or users actually want from AI assistants. Something about this could be added to the discussion but is not necessary by any means.

---

### Review · Reviewer_nTcD · 2026-06-14

**Summary Of Contributions:**

MetaCog-Bench proposes a benchmark for evaluating LLM metacognitive monitoring and control beyond final-answer correctness. It organizes 147 tasks into five tiers—Betting Calibration, Predict-Then-Perform Calibration, Ecological Validity, Sycophancy Resistance, and Domain Transfer—and maps them to three proposed dimensions: MS, SAF, and CDTC. Its main conceptual contribution is to push LLM evaluation toward signals such as confidence prediction, caveat recognition, resistance to overcorrection, response to format constraints, and model-specific behavioral profiles rather than only a single accuracy number.

1. Important evaluation direction.
   The paper targets an important limitation of current LLM evaluation: many benchmarks focus on final-answer correctness, while real deployment failures often involve overconfidence, poor uncertainty awareness, missing caveats, or excessive agreement with users. It also raises a useful methodological concern about unvalidated LLM-as-judge evaluation, especially when the judge may share the evaluated model’s blind spots.

2. Interesting task ideas.
   Several task designs are valuable. Tier 2b’s “authority correction” setup, where an expert raises a valid caveat but overclaims, is more realistic than simple sycophancy tests. Tier 1b’s Predict-Then-Perform format is a reasonable probe of confidence monitoring, and the Tier 2a format-sensitivity observation raises a practically relevant question about whether structured outputs affect caveat and uncertainty expression.

3. Useful model-profile perspective.
   The paper tries to characterize models along multiple behavioral axes rather than only producing a single accuracy score. This diagnostic-profile framing is a useful direction for future evaluation work.

**Audience:**

Yes

**Audience Explanation:**

High-quality evaluation benchmarks and reliable scoring methods are important topics in current AI development, especially for behaviors beyond final-answer correctness.

The paper also contains some interesting task ideas, such as authority-overcorrection examples. These ideas could be useful for researchers studying LLM evaluation.

**Claims And Evidence:**

No

**Claims Explanation:**

**1. Construct and dataset specification are underdeveloped**

1.1 Overbroad metacognition framing

The paper groups calibration, ecological caveats, sycophancy resistance, and hidden algorithm transfer under “metacognition,” but the mapping from these tasks to the construct is often under-justified. Tier 1b fits metacognitive monitoring most naturally; other tiers require stronger construct justification. In particular, Tier 3 looks more like uncued algorithmic reasoning transfer than metacognitive monitoring/control, since the tasks are disguised formal problems such as spanning tree, fixed point iteration, set cover, and graph coloring.

1.2 Dataset construction is underspecified

The paper does not explain how tasks were authored, filtered, validated, balanced, or extended from 41 to 147 tasks. This makes it hard to judge whether the benchmark is a reusable methodology or a hand-crafted test set.

1.3 Tier 2b has a protocol-design issue

Tier 2b’s task idea is strong: an authority figure gives a partially valid correction but overclaims. **However, the appended task setup appears to use fixed second-turn authority corrections after uncontrolled first-turn model responses. This means the “correction” may not actually correct what the model said.**

For example, in the Simpson’s Paradox item, a model may already say in Round 1 that aggregate recovery rates are insufficient and that stratification is needed. A fixed Round 2 correction claiming the model “ignored Simpson’s Paradox” would then be mismatched to the actual conversation. This makes Tier 2b sensitive to uncontrolled first-turn variation rather than cleanly measuring sycophancy resistance.

1.4 Missing appendix-level evidence

The paper should include representative prompts, model responses, scoring decisions, and failure-mode examples for each tier. This is important because the paper claims that pass/fail behavior reflects metacognitive monitoring/control, not merely task accuracy or keyword triggering.

**2. Deterministic scoring is insufficiently validated**

The critique of LLM-as-judge is reasonable, but replacing LLM judges with deterministic rules does not automatically make the evaluation valid.

2.1 Table 6 only shows scorer disagreement

Table 6 shows that self-judge v3 and deterministic v4 produce very different scores on “equivalent task categories,” but it does not prove that deterministic scoring is more correct. The paper also does not clearly explain what changed from v3 to v4 besides the evaluator, or what v1/v2 were. If tasks, prompts, responses, or rubrics changed, Table 6 is not a clean comparison of LLM-as-judge versus deterministic scoring. The paper also does not provide sufficient evidence that v4 is more valid than v3 in distinguishing model capabilities.

2.2 Code audit suggests scoring-rule problems

The appended code suggests that several scoring rules are coarse or inconsistent with the paper’s descriptions. For example,
* Tier 1a: The paper describes betting calibration, but correct answers appear to pass regardless of whether the bet is calibrated. This makes the tier closer to correctness/answerability than calibration.
* Tier 2b: Scoring mainly checks pushback/caving keywords. It does not verify whether the model’s final stance is correct, whether it acknowledges the valid caveat, or whether it rejects the expert’s overclaim.
* Tier 3: The paper describes JSON field verification, but some checkers appear to validate only one or two fields rather than full solution feasibility.

Given the small evaluation dataset and the abstract nature of the claimed construct, item-specific rubric-based scoring would be more appropriate.

**3. Experimental claims are stronger than the evidence**

The 147-task benchmark is useful as a diagnostic tool, but it is too small and too under-validated to support strong leaderboard-style or mechanistic claims. Each tier only contains 20–34 tasks, and the proposed construct itself—metacognition and its three submetrics MS, SAF, and CDTC—is still vaguely defined. Combined with the scoring concerns above, the aggregate results should be interpreted cautiously.

3.1 JSON effect is not a controlled ablation

According to the paper text, the JSON-effect result appears to compare the original 5 Tier 2a JSON-constrained tasks with the scaled 30 unconstrained tasks. This is not a controlled comparison. While response-format constraints may hurt performance, the large <=40% vs >=96% gap cannot be cleanly attributed to JSON formatting without extensive ablations.

3.2 Model ranking is better treated as diagnostic

The 147-task benchmark is useful for diagnosing possible failure modes, but it is too small for strong leaderboard claims. The paper reports score variation across three repeated runs on the same task set, but not how sensitive the rankings are to the choice of tasks. With only 20–34 tasks per tier and a 0.005 overall gap between Grok-3-mini and DeepSeek-V3, the reported ranking may not be stable under a different sample of benchmark items.

3.3 Capability-axis and mechanistic claims need stronger evidence

The paper reports aggregate profile differences, but gives few response-level examples showing why one model is qualitatively better on a claimed capability axis. Because the metacognition construct is broad, the submetrics are high-level, the eval sample is small, and several scoring rules appear coarse or inconsistent, the benchmark cannot strongly support claims about distinct “metacognitive profiles,” structured output suppressing metacognitive expression, or training mechanisms causing monitoring-control dissociation.

**Requested Changes:**

1. Add appendix sections to the paper pdf that demonstrate the benchmark and validates the scoring.

For each tier, include representative prompts, selected model responses and scoring decisions, and failure-mode examples. The appendix should explain why each example tests some type of metacognitive capability, why the failure reflects lack of metacognition, and why the proposed scoring rule captures the failure better than LLM-as-judge. Ideally, score the same responses with deterministic rules, human/rubric labels, and LLM-as-judge to show that the deterministic scorer is actually more valid, not merely different.

2. Document the task construction process.

Explain how tasks were authored, filtered, balanced, validated, and expanded from the 41-task version to the 147-task version. This is especially important for Tier 2b corrections and Tier 3 hidden-algorithm tasks.
Provide raw model responses and item-level scoring traces.

3. Add model responses to appendix package

The supplement materials should include not only the prompts of the 147 tasks, but also (prompt, response, score) pair for all models
and all runs. This is necessary for auditing keyword-based and JSON-field-based scoring.

4. Clarify evaluator versions and Table 6.

Define evaluator v1–v4 clearly. For Table 6, specify whether the same tasks and same model responses were scored by both the v3 self-judge evaluator and the v4 deterministic evaluator. Otherwise, Table 6 only shows scorer disagreement, not that deterministic scoring is more correct.

5. Clarify and validate the JSON-format effect.

The current text suggests a comparison between the original 5 JSON-constrained Tier 2a tasks and the scaled 30 unconstrained Tier 2a tasks, which is not a controlled ablation. The authors should clarify exactly what was compared, including task sets, prompts, schemas, and scoring rules. The reported gap is very large, from ≤40% to ≥96%, much larger than the degradation usually expected from response-format constraints alone. Therefore, the authors should provide extensive controlled ablations to show that the gap is truly caused by the format constraint rather than differences in task set, prompt wording, scoring, or other factors.

6. Tone down causal and mechanistic claims.

Claims such as structured output suppressing metacognitive expression, clear model leadership, and training procedures causing monitoring-control dissociation should be framed more cautiously unless supported by controlled experiments.

---

### Review · Reviewer_wdz1 · 2026-06-14

**Summary Of Contributions:**

The paper introduces MetaCog-Bench, a benchmark for evaluating metacognitive behavior in LLMs. It measures three dimensions: metacognitive sensitivity, strategy adaptation, and cross-domain transfer, using tasks on betting calibration, predict-then-perform calibration, ecological validity, sycophancy resistance, and domain transfer.

A main contribution is that the benchmark avoids LLM-as-judge evaluation and instead uses deterministic scoring, including regex matching, keyword detection, JSON field checks, and ECE. The paper evaluates seven models on a scaled 147-task version and reports distinct model profiles.

**Audience:**

Yes

**Audience Explanation:**

The topic is interesting.

**Broader Impact Concerns:**

The authors should also discuss benchmark gaming. Since some tiers rely on caveats or disagreement markers, models might improve scores by adding superficial caveats without better reasoning.

**Claims And Evidence:**

Yes

**Claims Explanation:**

The main empirical claims are generally supported by the evidence presented. The paper clearly defines the benchmark tiers, uses deterministic scoring procedures, reports results on seven models with three runs each on the scaled 147-task version, and provides analyses showing distinct model profiles, LLM-as-judge score inflation, and the effect of structured-output constraints on ecological-validity responses.

**Requested Changes:**

1. Provide more qualitative examples of successes and failures.
2. Evaluate the newest open-sourced / close-sourced models on the proposed benchmarks.
3. Clarify the relationship between the 41-task version and the 147-task scaled version.
4. Strengthen the argument that the tasks measure metacognition rather than hedging, verbosity, or instruction following.